# JumpStyle: Jump Starting Style Aware Test-Time Domain Generalization

## Abstract

The performance of deep networks is quite vulnerable to distribution shifts encountered during test-time, and this applies even for models which have been trained to generalize to unseen domains. Thus, it is imperative that the model updates itself, leveraging the test data in an online manner. In this work, we propose a novel framework for test-time adaptation of deep networks trained in the Domain Generalization (DG) setting. Specifically, we propose two modifications, (i) Jump starting the adaptation using effective initialization and (ii) Style-aware augmentation based pseudo-labelling over the current state-of-the-art approach for test-time adaptation, namely Tent, for the DG task. The proposed framework only assumes access to the trained backbone and is agnostic to the model training process. We demonstrate the effectiveness of the proposed JumpStyle framework on four DG benchmark datasets, namely, PACS, VLCS, Office-Home and Terra-Incognita. Extensive experiments using standard backbones trained using multiple source domains and also the state-of-the-art DG method shows that the proposed framework is generalizable not only across different backbones, but also different training methods.

## 1 Introduction

Research in deep learning has achieved remarkable progress in solving several computer vision tasks like image classification, object detection, segmentation Deng et al. (2009); Lin et al. (2014); Everingham et al. (2010); Chen et al. (2017); He et al. (2017); Ren et al. (2015). However, their performance usually drops significantly when data from previously unseen domains are encountered during testing, which is quite common in real scenarios. To overcome this, there has been a considerable amount of research interest in areas like Unsupervised Domain Adaptation (UDA), Domain Generalization (DG), etc. UDA setting assumes access to labelled source domain data and unlabelled target domain data during training. However, the target domain samples may not be available during model training. This is addressed in the DG setup where the objective is to use multiple source domains to learn domain invariant representations, thus preparing the model for future deployment, where the test samples can come from a different domain. But, none of these approaches leverage the rich information inherently present in the test data, which may help to improve the test performance.

Motivated by this, recently, Test-Time Adaptation (TTA) methods are gaining increasing importance, as they can leverage the testing data to reduce the adverse effect of distribution shift during testing. Here, we specifically focus on the approaches which can adapt any off-the-shelf model using the unlabeled test data in an online fashion. Very recently, a classifier adjustment framework Iwasawa & Matsuo (2021) was proposed for test-time adaptation in DG setup, which reports impressive performance for several backbones. Inspired by this work, here, we present a complementary analysis, where we analyze if different backbones trained using simple Empirical Risk Minimization (ERM) or even state-of-the-art DG approaches Zhou et al. (2021) specialized for generalization can further benefit from TTA using unlabelled test data.

Towards this goal, we build upon the state-of-the-art TTA method Tent Wang et al. (2021) and suitably adapt it for the DG application. Our proposed simple yet effective framework termed **JumpStyle**, consists of two main components: (i) **Jump** start initialization that updates Batch-Normalization (BN) statistics as

a convex combination of the source as well as the test data statistics, conditioning the mixing coefficient on the number of test samples available. (ii) Consistency of **Style**-aware augmentations for pseudo-labelling the unseen target domain data for updating the BN affine (scale and shift) parameters, in addition to the objective of test entropy minimization as in Tent.

Our contributions can thus be summarized as follows:

- We propose a novel framework, namely *JumpStyle* for addressing the task of test-time adaptation in the Domain Generalization setup.

- JumpStyle can be seamlessly integrated with several backbones trained using various DG approaches.

- Extensive experiments on four benchmark datasets using different backbones, also trained using different DG methods demonstrate the effectiveness of the proposed framework.

## 2 Related Work

Here, we briefly review some recent literature relevant to our work.

### 2.1 Domain Generalization

The objective of DG is to learn domain invariant representations using labelled data from multiple source domains belonging to the same classes, such that the trained model is robust to unseen test domains. In Li et al. (2018b), adversarial autoencoders are used to learn generalized features by minimizing the Maximum Mean Discrepancy (MMD) measure. An auxiliary unsupervised learning task of solving jigsaw puzzles along with the classification task to discover invariances and regularities was proposed in Carlucci et al. (2019). Another line of work Li et al. (2018a); Balaji et al. (2018) uses meta-learning to synthesize pseudo-training and test domains to generalize to domain shifts during testing. The Invariant Risk Minimization (IRM) learning paradigm proposed in Arjovsky et al. (2019), estimates invariant, causal predictors from multiple domains, enabling generalization to unseen domains. In Kim et al. (2021), a self-supervised contrastive regularization method was proposed, while in Li et al. (2021), the features are perturbed with Gaussian noise during training. In Robey et al. (2021), the DG problem is formulated as an infinite-dimensional constrained statistical learning problem, for which a novel algorithm inspired from non-convex duality theory was proposed. Recently, Zhou et al. (2021) proposes to mix instance level feature statistics implicitly to synthesize pseudo domains. This increases the diversity of source domains resulting in a model with better generalization ability.

### 2.2 Test Time Adaptation

TTA refers to adapting the trained model during testing to improve the performance on the test data, which can come from an unseen domain. Although the test data is unlabeled, they provide rich domain information which can be utilized to update the trained model during deployment. Due to its wide applicability for real world deployment of deep networks, this field is gaining increasing attention. In Schneider et al. (2020), they propose to update BN statistics as a convex combination of the previously estimated training data statistics and the test data statistics to mitigate effect of domain shift. Test-Time-Training (Sun et al., 2020) introduces a self-supervised task along with that of classification during training and while testing, the model is adapted using just the self supervision objective. However, fully test time adaptation is done independent of the training phase, given a trained model and unlabeled test data. This was addressed in Tent (Wang et al., 2021), where they propose to update BN affine parameters to minimize the test entropy, enforcing confident model predictions. The above approaches were evaluated where the model was trained using a single source domain, and was tested on samples from a different domain. Only recently, a TTA approach for the DG setup was proposed, namely T3A (Iwasawa & Matsuo, 2021). It is an optimization free adaptation method, where the trained linear classifier is adjusted using the online available unlabelled test data.

## 3 Problem Definition

Here, we first explain the task of domain generalization and then test-time adaptation.

**Domain Generalization (DG):** The goal of DG is to train a model with labeled data from multiple domains, such that it can generalize well to unseen domains during testing. The training data comprises of multiple source domains, i.e., $\mathcal{D}_{train} = D_1 \cup D_2 \ldots \cup D_{d_{tr}}$, where each source domain consists of labelled samples, which we denote as $D_d = \{(x_i^d, y_i^d), i = 1 \ldots n_d\}, d = 1, \ldots, d_{tr}$. Here, $n_d$ denotes the number of samples in domain $d$ and $d_{tr}$ denotes the number of training domains. The objective is to learn a model $F_{\boldsymbol{\theta}}$ using $\mathcal{D}_{train}$, such that it can generalize well to an unseen test domain $D_{test} \notin \mathcal{D}_{train}$. Here, $\boldsymbol{\theta}$ denotes the parameters of the trained model.

**Testing phase:** For the standard DG task, the trained model $F_{\boldsymbol{\theta}}$ is directly used for testing. Since the model is trained using data from multiple source domains to compute domain-invariant representations, it is expected to work well for unseen domains during testing. But the test data also contains rich information about the domain, which is usually not leveraged by these algorithms to further improve their performance. In general, during testing, the data is available in batches in an online manner, and thus this data can be used for TTA of the trained model $F_{\boldsymbol{\theta}}$ to further improve the performance.

## 4 Baseline Approaches

The performance of DG models on the test data depends significantly on the training process. If the training is not effective, thereby resulting in poor generalizability, TTA may help in improving the performance on the unseen domain test samples. In contrast, for well-trained DG models which generalize well to unseen domains, it is not obvious whether TTA can help to further improve their performance. Needless to say, the performance also depends on the backbone architecture. Based on this intuition, for this work, we choose two DG baselines which we discuss below.

**(1) Empirical risk minimization (ERM)** (Vapnik, 1998): A simple, yet strong baseline in DG setup is to use the multiple source domain samples together to train the network $F_{\boldsymbol{\theta}}$ by minimizing the following empirical risk:

$$\mathcal{L}_{ERM} = \frac{1}{d_{tr}} \sum_{d=1}^{d_{tr}} \frac{1}{n_d} \sum_{i=1}^{n_d} \mathcal{L}_{CE}(x_i^d, y_i^d) \tag{1}$$

Here, $(x_i^d, y_i^d)$ refers to the $i^{th}$ labelled sample from domain $d$ and $CE$ refers to the standard Cross-Entropy loss. We experiment with two different backbones to analyze their test-time adaptation performance in the DG scenario.

**(2) MixStyle** (Zhou et al., 2021): As the second baseline, we choose the recent state-of-the-art DG approach, namely MixStyle. Here, pseudo-domains are synthesized by mixing feature statistics of two different instances. These label preserving feature perturbations help to regularize the CNN, thereby learning class discriminant, yet domain invariant features. The perturbed features are then used to minimize the CE loss as in eqn. (1). Generating pseudo-domains/styles helps to achieve excellent performance for unseen domain examples during testing.

In this work, we develop a simple, yet effective framework for updating the DG model in an online fashion during test time. We analyze whether the proposed framework can improve upon DG models using different backbones as well as trained using different techniques. Here, all the baselines including MixStyle Zhou et al. (2021), use BN layers. Thus, this work will serve as a complementary analysis to the state-of-the-art T3A framework Iwasawa & Matsuo (2021), which analyses DG backbones without BN layers.

## 5 Proposed Method

The proposed JumpStyle framework for test-time DG, has two main modules, (i) Effective initialization (Jump Start) and (ii) Consistency of style-aware augmentations for pseudo-labelling. It is built upon the successful TTA method Tent (Wang et al., 2021), which we describe below.

**Entropy Based Tent Framework:** Tent is a fully test-time adaptation method designed to adapt any given off-the-shelf model using only the available test data, which can come from a distribution different from that of the training data. In general, during training, the BN layers estimate the channel-wise statistics $\mu, \sigma$ of the feature maps using the training data. While these statistics are relevant when the test samples are drawn from the same distribution as the training data, they are not optimal when there is a distribution shift during testing. This is because the feature maps would no longer be normalized to zero mean and unit variance, a phenomenon known as covariate shift. In Tent, the BN statistics of the trained model are replaced with that of the test data. Specifically, given features $f$ at a certain layer, in general, BN normalizes the feature as $\hat{f} = (f - \mu_s)/\sigma_s$ and performs affine transformation as $f_{BN} = \gamma \hat{f} + \beta$. In Tent, instead of the source data statistics $\{\mu_s, \sigma_s\}$, the test data statistics $\{\mu_t, \sigma_t\}$ are used. Further, the BN affine parameters $\{\gamma, \beta\}$ are finetuned to minimize the test prediction entropy (defined later).

Before describing the proposed JumpStyle framework, we explain the reasons behind choosing Tent as the base approach in our work: (i) Tent can use any off-the-shelf trained model and does not assume any knowledge about the method of training; (ii) For the DG models with BN layers, the performance of Tent is comparable to the state-of-the-art T3A for some domains (Iwasawa & Matsuo, 2021); (iii) This will be a complementary analysis to that in T3A as our work focuses on backbones with BN layers and trained using different DG methods, whereas T3A mainly focuses on backbones without BN layers; (iv) Instead of proposing a completely different approach targeted towards test-time DG, we want to explore whether advances in the field of test-time adaptation can be made useful for related tasks.

We now describe the two proposed modifications of the Entropy-based Tent framework for this application. Specifically, given a trained model $F_{\boldsymbol{\theta}}$ parameterized by $\boldsymbol{\theta}$, our objective is to utilize the target samples $x_t$ of batch size $n$, to adapt the model.

**1) Jump start initialization of the BN parameters:** Tent (Wang et al., 2021) accounts for covariate shift by substituting the training BN statistics with those of the test data . But during online testing, the number of target samples available at any stage is usually quite less and variable, and thus may not be a good representative of the entire target distribution. Here, we propose a simple, yet effective correction of the test batch statistics by utilizing the training domain statistics as the prior to improve the performance under domain shift (Schneider et al., 2020). Since the quality of the estimated test domain statistics depends on the test batch size $n$, the source and target statistics are combined as follows

$$\bar{\mu} = \alpha(n)\mu_s + (1 - \alpha(n))\mu_t$$
$$\bar{\sigma^2} = \alpha(n)\sigma_s^2 + (1 - \alpha(n))\sigma_t^2 \tag{2}$$

where $\mu_t$ and $\sigma_t^2$ are the online estimated test distribution statistics, $\mu_s$ and $\sigma_s^2$ are the source data statistics available as part of the given trained model.

The weight $\alpha(n)$ is a function of batch size $n$ and has a significant effect on the final performance. In Schneider et al. (2020), a method was proposed to compute this weight based on the batch size $n$ and an additional hyper-parameter. Since the weight should ideally be a function of only the batch-size, in this work, we design $\alpha(n)$ to be:

$$\alpha(n) = 0.5(1 + e^{-\kappa n}); \quad \text{where } \kappa = 0.05 \tag{3}$$

The weight is designed such that it satisfies the following criteria: As the number of samples in the batch $n$ decreases, the weight for the source statistics $\alpha(n)$ should increase. In the extreme case, when $n = 0$, $\alpha(n) = 1$. But when $n > 0$, since the number of test samples available is still limited, the smallest value of $\alpha(n)$ is constrained to not fall below 0.5. The value of $\kappa$ is obtained empirically, but the proposed weighting rule has the advantage that it only depends on the batch-size as desired. This weighting is used for all the experiments reported in this work. In addition to this initialization, after the data from a test-batch is passed through the model, its style-aware weak and strong augmentations are used to further update the BN affine parameters.

**2) Pseudo-labelling based on consistency of style-augmented targets:** During testing, as the test batches become available in an online manner, samples with confident predictions can be pseudo labelled and used to supervise the model for target data. Pseudo labels obtained with the criteria of consistent

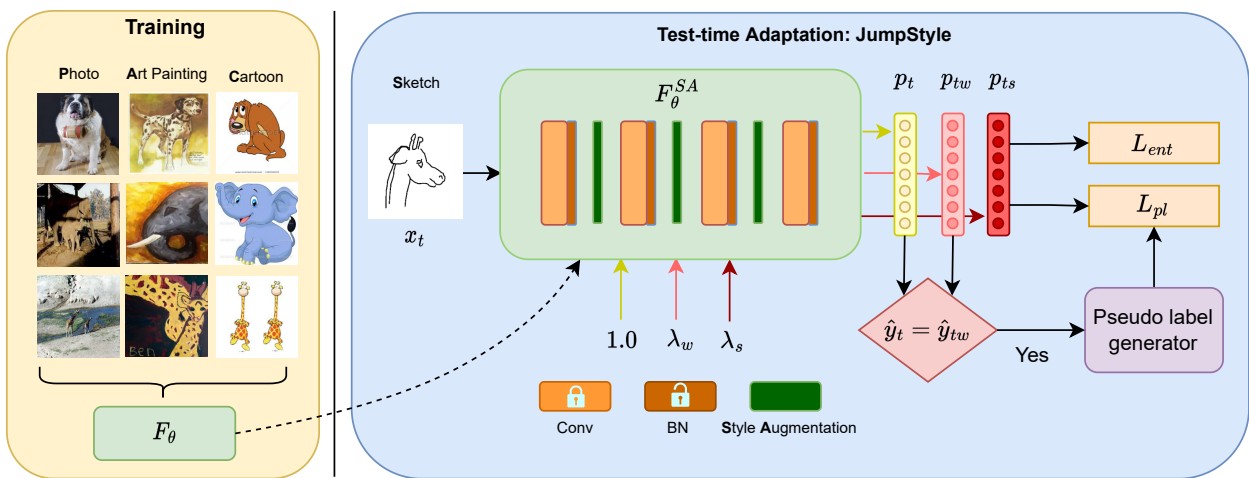

Figure 1: DG training (left) using Photo, Art-painting, Cartoon as source domains. TTA using JumpStyle (right) on test sample $x_t$ from test domain *Sketch*. Consistency across predictions of true sample $p_t$ and weak style augmentation $p_{tw}$ are used to pseudo label $x_t$. BN affine parameters are updated to minimize the pseudo label and entropy loss.

predictions across augmented versions of unlabelled data has shown remarkable success in semi-supervised learning (SSL) Sohn et al. (2020); Berthelot et al. (2019). Here, we explore whether pseudo-labelling based on suitable consistency condition aids test-time DG scenario, which to the best of our knowledge, has not been explored before. We propose to check consistency of style-augmented target samples when they are available, which is more suited for the DG task. The goal of DG is to generate domain (style used interchangeably)-invariant representations. Thus, the model should be able to consistently predict the same class label for a sample and its augmentations, with the same content, but different styles.

During testing, given two target samples $x_i$ and $x_j$, we create an augmented feature of $x_i$ using the style of $x_j$. Let $f_i$ and $f_j$ denote their respective feature maps at a certain layer. The channel-wise means $\mu(f_i), \mu(f_j)$ and standard deviations $\sigma(f_i), \sigma(f_j)$ are representative of the image styles. In this layer, these feature statistics are mixed, thereby generating a pseudo-style, which is then applied to the style normalized feature of $f_i$ to obtain style augmented feature $f_i^{SA}$ Zhou et al. (2021)

$$
\begin{aligned}
\mu_{mix}(f_i; \lambda) &= \lambda\mu(f_i) + (1-\lambda)\mu(f_j) \\
\sigma_{mix}(f_i; \lambda) &= \lambda\sigma(f_i) + (1-\lambda)\sigma(f_j) \\
f_i^{SA} &= \sigma_{mix}(f_i; \lambda) * \frac{f_i - \mu(f_i)}{\sigma(f_i)} + \mu_{mix}(f_i; \lambda)
\end{aligned}
\tag{4}
$$

where $\lambda \in [0, 1]$ is the mixing coefficient. Features thus obtained preserve the semantic content of the input $x_i$, while only the style is perturbed using that of the other image.

Inspired by Sohn et al. (2020), we compute two types of style augmentations for each target sample, namely weak style augmentation and strong style augmentation as described next. Let $F_{\boldsymbol{\theta}}^{SA}(; \lambda)$ denote the entire model including the feature extractor, classifier and the softmax layer with the style augmentations. Setting the mixing coefficient $\lambda = 1$ reduces the model $F_{\boldsymbol{\theta}}^{SA}(; \lambda)$ to the original backbone $F_{\boldsymbol{\theta}}$. Given a test batch $x_t$, the samples are randomly permuted within the batch to obtain $\tilde{x}$. The features of $x_t$ are perturbed by instance-wise mixing of styles from features of $\tilde{x}$ as described in eqn. (4). For a sample $x_t$, we denote its prediction as $p_t$, and those of its weak and strong augmentations as $p_{tw}$ and $p_{ts}$ respectively. These are obtained as follows

$$
p_t = F_{\boldsymbol{\theta}}^{SA}(x_t; 1); \quad p_{tw} = F_{\boldsymbol{\theta}}^{SA}(x_t; \lambda_w); \quad p_{ts} = F_{\boldsymbol{\theta}}^{SA}(x_t; \lambda_s)
\tag{5}
$$

To better utilise the target samples during test-time, we generate pseudo labels for the samples whose predictions are confident and robust against weak domain shifts. The pseudo labels for the test sample and

| Backbone | Method | VLCS | PACS | OfficeHome | Terra | Average |
|---|---|---|---|---|---|---|
| ResNet-50 | ResNet-50 | 74.3±0.5 | 84.1±0.1 | 66.9±0.2 | 45.8±1.8 | 67.8 |
| | SHOT-IM | 61.5±1.7 | 84.6±0.3 | 68.0±0.0 | 33.8±0.3 | 62.0 |
| | SHOT | 61.6±1.8 | 84.8±0.5 | 68.0±0.0 | 34.6±0.3 | 62.3 |
| | PL | 63.4±1.8 | 80.1±3.5 | 61.3±1.5 | 36.8±4.4 | 60.4 |
| | PL-C | 73.3±0.8 | 84.7±0.3 | 66.4±0.3 | **47.0±1.7** | 67.9 |
| | Tent-Full | 75.4±0.6 | 87.0±0.2 | 66.9±0.2 | 42.6±0.8 | 68.0 |
| | BN-Norm | 71.3±0.4 | 85.8±0.1 | 66.4±0.1 | 42.3±0.4 | 66.5 |
| | Tent-C | 72.4±1.5 | 84.4±0.1 | 66.2±0.2 | 42.4±3.1 | 66.4 |
| | Tent-BN | 65.6±1.4 | 84.9±0.0 | 67.7±0.2 | 42.7±0.5 | 65.2 |
| | T3A | 76.0±0.3 | 85.1±0.2 | 68.2±0.1 | 44.6±0.9 | 68.5 |
| | **JumpStyle** | **76.9±0.7** | **87.5±0.6** | **69.1±0.5** | 44.7±0.7 | **69.5** |
| ResNet-18 | ResNet-18 | 73.0±0.6 | 79.5±0.4 | 61.8±0.3 | 41.7±0.9 | 64.0 |
| | SHOT-IM | 61.6±0.3 | 82.1±0.3 | 62.5±0.3 | 32.8±0.4 | 59.8 |
| | SHOT | 61.8±0.3 | 82.3±0.2 | 62.8±0.2 | 32.7±0.4 | 59.9 |
| | PL | 67.0±0.6 | 72.9±1.0 | 56.3±2.5 | 35.4±1.7 | 57.9 |
| | PL-C | 71.8±1.3 | 78.9±0.4 | 61.7±0.3 | **43.1±0.9** | 63.9 |
| | Tent-Full | 72.3±0.3 | 83.9±0.3 | 62.7±0.2 | 36.9±0.3 | 64.0 |
| | BN-Norm | 70.4±1.0 | 82.7±0.1 | 62.0±0.1 | 36.4±0.2 | 62.9 |
| | Tent-C | 71.3±1.5 | 74.6±1.9 | 60.5±0.4 | 40.9±0.5 | 61.8 |
| | Tent-BN | 64.7±0.7 | 81.1±0.2 | 62.5±0.3 | 36.4±0.9 | 61.2 |
| | T3A | 74.5±0.9 | 81.4±0.2 | **63.2±0.4** | 39.5±0.3 | 64.6 |
| | **JumpStyle** | **75.7±0.4** | **86.1±0.6** | **63.3±0.3** | 40.5±0.5 | **66.4** |

Table 1: Results with ERM approach using ResNet-50 and ResNet-18 backbones.

its weak augmentation are obtained as $\hat{y}_t = argmax(p_t)$ and $\hat{y}_{tw} = argmax(p_{tw})$ respectively. The pseudo label loss is then computed as

$$\mathcal{L}_{pl} = \mathbb{E}_{x_t \in \mathcal{S}}[-\log p_{ts}(\hat{y}_t)]; \qquad \mathcal{S} = \{x_t | \hat{y}_t = \hat{y}_{tw}; max(p_t) > \tau\} \tag{6}$$

Inspired from Tent (Wang et al., 2021), we also use entropy loss to enforce confident predictions. In this work, we define this only for the strong style augmentations as follows:

$$\mathcal{L}_{ent} = -\frac{1}{n}\sum_{t=1}^{n}\sum_{c} p_{ts}(c)\log p_{ts}(c) \tag{7}$$

where $c$ denotes the class index and $n$ is the test batch size.

Although inspired from the SSL approach Sohn et al. (2020), there are significant differences between the two approaches as: (i) The weak and strong style augmentations proposed in this work are better suited for the Domain Generalization objective as compared to the standard image augmentations as in Sohn et al. (2020) (details in experimental section). (ii) Unlike the semi-supervised approaches, where the whole network is trained/fine-tuned using the pseudo-labelling loss, here only the BN layers are updated.

**Final Test-time adaptation loss:** The total loss for adaptation during test time is computed as a weighted combination of the pseudo-label loss and the entropy loss. The BN affine parameters, denoted by $\{\boldsymbol{\gamma}, \boldsymbol{\beta}\}$ are updated in an online fashion each time a new batch is available, to minimize the following test time loss:

$$\mathcal{L}_{test} = \eta * \mathcal{L}_{pl} + (1 - \eta) * \mathcal{L}_{ent} \tag{8}$$

The parameter $\eta$ balances the two losses, and is empirically set to 0.8 for all the experiments.

| Method | PACS | | | VLCS | | |
|---|---|---|---|---|---|---|
| | Batch size=8 | Batch size=32 | Batch size=64 | Batch size=8 | Batch size=32 | Batch size=64 |
| MixStyle* | 82.4 | 82.4 | 82.4 | 75.7 | 75.7 | 75.7 |
| Tent-Full | 81.2 ±0.9 | 85.9±0.7 | 86.7±0.8 | 69.2±0.6 | 71.4±0.4 | 71.9±0.4 |
| T3A | 80.5±0.7 | 84.9±0.5 | 85.6±0.5 | 72.6±0.7 | 75±0.5 | 75.5±0.6 |
| **JumpStyle** | **85.9±0.7** | **86.4±0.6** | **86.8±0.6** | **76.3±0.4** | **76.5±0.3** | **76.1±0.3** |

Table 2: Results on PACS and VLCS datasets using MixStyle trained DG model. * denotes results obtained using the official MixStyle Zhou et al. (2021) implementation.

| Method | VOC | LabelMe | Caltech | SUN09 | Average |
|---|---|---|---|---|---|
| Tent | 69.2±0.3 | 60.6±0.5 | 91.0±0.8 | 66.0±0.6 | 71.7 |
| +Jump | 69.6±0.5 | 66.0±0.4 | 96.3±0.6 | 66.6±0.5 | 74.6 |
| +Jump+FixMatch | 69.8±0.5 | 64.8±0.3 | 95.8±0.3 | 67.7±0.4 | 74.5 |
| **+JumpStyle** | **71.3±0.4** | **66.5±0.5** | **96.5±0.7** | **68.5±0.7** | **75.7** |

Table 3: Ablation study on VLCS dataset using ResNet-18 backbone.

# 6 Experimental Evaluation

Here, we describe the experiments done to evaluate the effectiveness of the proposed framework. We perform experiments on four benchmark DG datasets demonstrating different types of domain shifts. **PACS** (Li et al., 2017) consists of four domains, Photo, Art painting, Cartoon and Sketch, where the domain shift is particularly due to image styles. It has $9,991$ images belonging to 7 classes. **VLCS** (Fang et al., 2013) is a collection of four datasets, Caltech101 (Fei-Fei et al., 2006), LabelMe (Russell et al., 2008), SUN09 (Choi et al., 2010), VOC2007 (Everingham et al., 2010) with $10,729$ samples from 5 classes. **Office-Home** (Venkateswara et al., 2017) consists of four domains, Art, Clipart, Product, Real-world, with $15,500$ images of 65 objects in office and home environments. **Terra-Incognita** (Beery et al., 2018) contains photos of wild animals. Following Gulrajani & Lopez-Paz (2021); Iwasawa & Matsuo (2021), we use the images captured at locations $L100, L46, L43, L38$ as the four domains. This contains 24788 examples of 10 different classes.

## 6.1 TTA-Baselines and Implementation details:

We compare the proposed JumpStyle with the following test-time adaptation baselines: 1) **SHOT-IM** Liang et al. (2020): updates the feature extractor to minimize entropy and the diversity regularizer; 2) **SHOT** Liang et al. (2020): uses pseudo-label loss along with information maximization as in (1); 3) **PL (Pseudo labelling)** Lee (2013): updates the entire network by minimizing the cross-entropy between the prediction and pseudo labels; 4) **PL-C** Lee (2013): minimizes the pseudo-label loss as above and updates only the linear classifier; 5) **Tent-Full** Wang et al. (2021): is the original method, where the BN statistics and transformations are updated; 6) **BN-Norm** Schneider et al. (2020): only the BN statistics are updated while keeping the affine parameters fixed; 7) **Tent-C** Wang et al. (2021): updates only the classifier to reduce the prediction entropy; 8) **Tent-BN** Wang et al. (2021): adds one BN layer just before the linear classifier and then modulates its affine parameters.

**Implementation Details:** Following Iwasawa & Matsuo (2021), we split the data in each domain into a training (80%) and validation (20%) split. We follow the leave one out protocol for training and evaluation. In each experiment, three domains act as the source whose training splits are used to train the model, while the validation splits are used to select the learning rate. Further, we perform test-time adaptation on the target domain and report the average accuracy over all the domains in the dataset. The parameters for a TTA framework has to be selected prior to deployment, before one has access to test data. Following T3A (Iwasawa & Matsuo, 2021), we set the batch size to 32 and use training domain validation

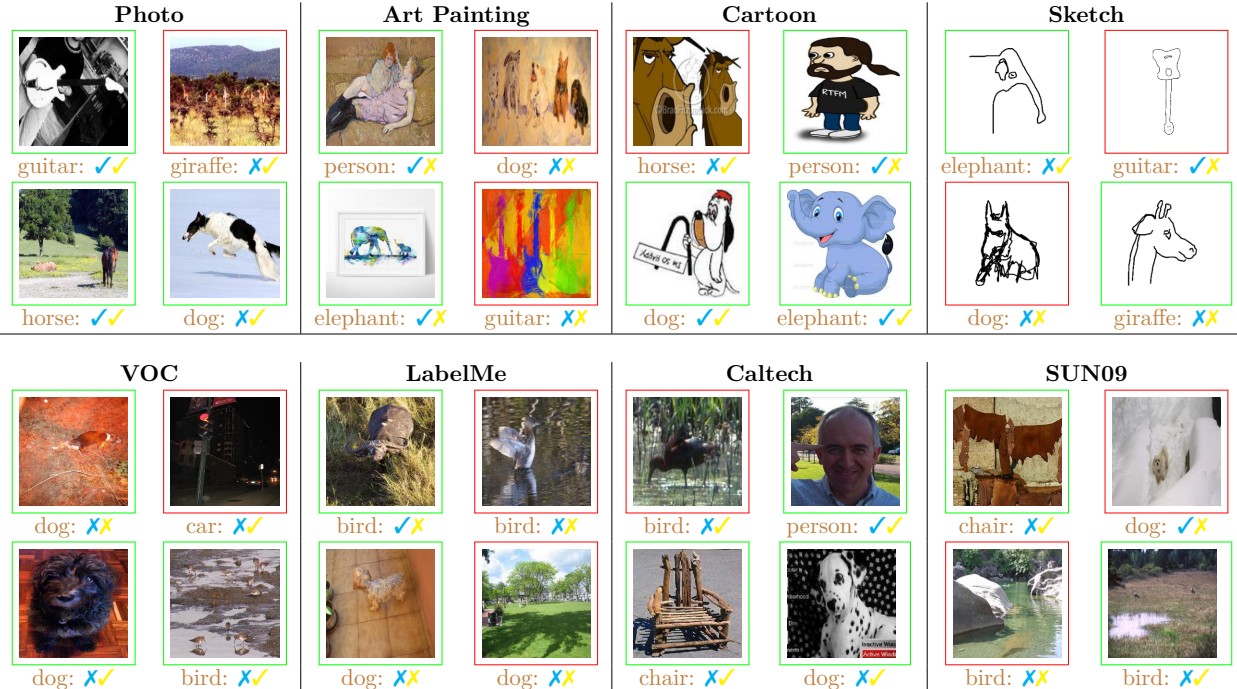

Table 4: Few examples from PACS and VLCS datasets where JumpStyle predicted the correct class (green box), and where it failed (red box), when compared with Tent (cyan) and T3A (yellow). The correct and incorrect predictions for T3A and Tent are marked with ✓and ✗respectively.

set to tune the hyperparameters for fair comparison. The learning rates used were $10^{-4}$ for PACS, VLCS, OfficeHome and $10^{-5}$ for Terra Incognita for both ResNet-18 and ResNet-50 backbones. We set $\alpha(n)$ to 0.6 which is computed using eqn.( 3) for n=32 and set $\eta$ to 0.8. We set $\lambda_w$ and $\lambda_s$ in eqn. (5) to 0.9 and 0.75 respectively. The parameter $\kappa$ in eqn. (3) is fixed to 0.05 for all the experiments.

## 6.2   Results with DG baselines:

**(1) Empirical Risk Minimization:** First, we test the proposed TTA framework with the ERM approach for DG, where labelled samples from multiple source domains are collectively used to train the network using CE loss. The results of the proposed framework and comparisons with the other approaches using ResNet-50 and ResNet-18 backbones are shown in Table 1 for the four datasets. The results of previous methods are directly taken from Iwasawa & Matsuo (2021). We observe that the proposed JumpStyle outperforms the other approaches for three of the four datasets, and also on an average. This explains the generalization ability of the proposed approach across different datasets and backbones.

**(2) Mixstyle:**   Here, we analyze whether TTA can also benefit from the state-of-the-art DG approaches, which have been designed specifically to obtain domain invariant representations. Since online TTA depends upon the test batch size, here, we also experiment with different batch sizes to analyze its effect on the final performance. We report the results obtained using MixStyle with ResNet-18 backbone and its performance on doing TTA using Tent-Full, T3A and JumpStyle in Table 2. From these results on PACS and VLCS datasets, we observe the following: (1) The performance of Tent-Full and T3A improves significantly for higher batch sizes. However, their performance is not satisfactory for smaller batch sizes. (2) The proposed framework outperforms all the previous approaches irrespective of the batch size. Table 4 shows the predictions of Tent, T3A and JumpStyle for a few examples from PACS and VLCS datasets. The proposed approach is indeed able to correct several samples which were wrongly predicted by previous TTA approaches.

### 6.3 Hyperparameter selection

As mentioned in Section 6.1, we use the training domains validation set to determine the hyperparameters $\eta$, $\alpha$ and the use of MixStyle layers.

| $\eta$ | VOC | LabelMe | Caltech | SUN09 | Average |
|---|---|---|---|---|---|
| 0.2 | 68.3±0.7 | 66.0±0.6 | 96.3±0.3 | 63.8±1.2 | 73.6 |
| 0.5 | 70.0±0.8 | 66.2±0.5 | 96.5±0.3 | 65.4±1.4 | 74.5 |
| **0.8** | **71.3±0.4** | **66.5±0.5** | **96.5±0.4** | **68.5±0.6** | **75.7** |

Table 5: Performance with varying $\eta$ on VLCS using ResNet-18.

1) We observed that $\eta = 0.8$ gave the best TTA performance on training domains validation set. For further insight, we vary $\eta$ in JumpStyle and report the results in Table 5. Higher $\eta$ implies higher weight for pseudo label loss when compared to entropy loss. Thus, consistency checked pseudo-labels provide stronger supervision and help to adapt to the target domain better, leading to improved performance.

| $\alpha(n)$ | VOC | LabelMe | Caltech | SUN09 | Average |
|---|---|---|---|---|---|
| 0.4 | 70.7±0.4 | 64.8±0.7 | 96.5±0.3 | 67.3±0.4 | 74.8 |
| 0.5 | 71.0±0.5 | 65.9±0.5 | 95.9±0.4 | 67.7±0.4 | 74.8 |
| 0.7 | 71.0±0.4 | 66.3±0.5 | 96.5±0.4 | 68.0±0.6 | 75.4 |
| **Ours (0.6)** | **71.3±0.4** | **66.5±0.5** | **96.5±0.4** | **68.5±0.6** | **75.7** |

Table 6: Performance with varying $\alpha$ on VLCS using ResNet-18.

2) We study the choice of $\alpha$ to mix the source and target BN statistics. As the batch size can be varying during test-time and the quality of test statistics depends on its(higher batch size gives better estimates), we perform experiments setting $\alpha$ to constants 0.4, 0.5, 0.7 and compare the results with the proposed choice of $\alpha(n)$ using eqn.(3).

| layers | VOC | LabelMe | Caltech | SUN09 | Average |
|---|---|---|---|---|---|
| 1 | 69.3±0.8 | 66.3±0.7 | 96.5±0.2 | 63.7±1.6 | 74.0 |
| 1, 2 | 70.33±0.7 | 66.4±0.5 | 96.3±0.3 | 65.8±1 | 74.7 |
| **1,2,3** | **71.3±0.4** | **66.5±0.5** | **96.5±0.4** | **68.5±0.6** | **75.7** |

Table 7: Performance with different layers for augmentation.

3)Based on the analysis presented in MixStyle (Zhou et al., 2021) and our experiments (Table 7), we insert the proposed Style Augmentation layers after the first three ResNet blocks as the early layers contain style information. Results in Table7 show that inserting these layers after each of the three ResNet blocks performs the best.

## 7 Conclusion

In this paper, we present a novel framework termed *JumpStyle* for test-time adaptation in domain generalization setup. Firstly, we propose an effective scheme to correct the BatchNorm statistics based on the number of test samples available online. Further, we propose a test-time consistency regularization method to ensure consistent predictions across perturbed versions of test samples. Unlike semi/unsupervised representation learning methods where augmentations in image space are observed to work effectively, our analysis shows that simple image augmentations are ineffective in the low-data test time scenario. We propose to augment the test samples in feature space instead. In specific, we use MixStyle which is a label preserving feature perturbation module to obtain weak and strong augmentations, across which we enforce consistent predictions. Extensive experiments performed using backbones with different representation ability, training methods and augmentations demonstrate the effectiveness of the proposed framework.

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
