# OpenReview forum: "JumpStyle: Jump Starting Style Aware Test-Time Domain Generalization"
_TMLR — Rejected by TMLR_

### Review · Reviewer_184g · 2022-12-02

**Summary Of Contributions:**

The paper proposes a test-time adaptation approach for domain generalization, which operates in an online manner and adapts to batches of testing examples. The proposed approach features two main components: (i) jump starting batch normalization by averaging the batch norm stats between training and test data depending on the batch size, (ii) style augmentations with pseudo-label loss and entropy minimization. The paper is evaluated on standard domain generalization datasets and is shown to outperform various baseline approaches.

**Audience:**

Yes

**Claims And Evidence:**

No

**Requested Changes:**

Questions:

1. [critical] Please elaborate the motivation of the online setup, assumptions on the testing distribution, and how it fits in the training procedure and experimental framework. Is there an assumption that the test data is from a single domain with predefined domain boundary? If so, why not use running averages as in batch norm training procedure to further improve the estimation of BN stats?

2. [critical] Ablation studies, or other types of evidence, are needed regarding why using weak augmentation to obtain pseudo-label and strong autmentation for entropy minimization? If the weak augmentation is only taking examples with high label agreement, wouldn’t it somewhat equivalent to entropy minimization, i.e., making the prediction more confident if it is already robust?

3. [non-critical] Ablation studies are needed to justify the need for style-augmentation over other types of input augmentation. The authors should also clarify the difference between styles and domains, and how style augmentation within the same test domain could be helpful.

4. [critical] In Sec. 6.3 (3), it is unclear why alpha is a constant hyper-parameter instead of a function over batch size. The authors should also clarify how alpha is configured in the ablation with different batch sizes in Table 2.

**Strengths And Weaknesses:**

Strengths

1. The area of test test domain adaptation in the online setting is an interesting area of research and the proposed batch size aware batch normalization is shown to improve the model’s robustness under low batch sizes.

2. The proposed approach demonstrates sota or competitive performance against various baseline approaches.

3. The proposed approach can be applied to different domain generalization algorithms such as ERM and Tent.


Weaknesses

1. Unclear problem formulation regarding the online learning setup: it is unclear what motivates the online setup of domain generalization, what are the underlying assumptions, and how it is validated in the experiments. The proposed approach is only evaluated on a single held-out test domain with predefined domain boundary, whereas in the typical online learning setup, the test data can be dynamic.

2. Unjustified complexity in loss function: the proposed loss function is a combination of various sota losses in semi-supervised learning, domain adaptation, test-time adaptation and domain generalization. Some of the loss details are not justified in the experiments: why use weak augmentation to obtain pseudo-label and strong autmentation for entropy minimization? If the weak augmentation is only taking examples with high label agreement, wouldn’t it somewhat equivalent to entropy minimization, i.e., making the prediction more confident if it is already robust?

3. Claims on style-augmentation is not supported by the experiments: if a single held-out domain is used in testing, it wouldn’t contain styles from other domains. It is unclear why style-augmentation is better than other types of input augmentation.

---

> ### Author Response · Authors · 2022-12-20
> **Response to Reviewer 184g**
>
> Dear reviewer,
>
> Thank you for the valuable feedback! Please find the detailed answers for the requested questions below:
>
> >Please elaborate the motivation of the online setup, assumptions on the testing distribution, and how it fits in the training procedure and experimental framework. Is there an assumption that the test data is from a single domain with predefined domain boundary? If so, why not use running averages as in batch norm training procedure to further improve the estimation of BN stats?
>
> The test-time adaptation setting for the domain generalization set-up is as follows:
> Training Stage: The model is first trained using multiple source domains as in the standard domain generalization scenario.
> Testing Stage: Given only the trained model, it is adapted in an online manner where the test samples come from an unseen test domain which is different from the source domains seen during training.
>
> In this work, as in T3A, we assume that the test data comes from a single domain.
>
> As suggested, we perform experiments on the VLCS dataset using the standard Batch norm training procedure which inherently updates the BN stats through running averages. All the other components of the proposed framework remain the same. We observe that the proposed JumpStart initialization in JumpStyle works significantly better compared to using running average for updating the batch norm parameters (referred as BN Standard).
>
> |    Method     |       VOC       |     LabelMe     |     Caltech     |      SUN09      |       Avg       |
> |:-------------:|:---------------:|:---------------:|:---------------:|:---------------:|:---------------:|
> |  BN standard  |   69.5 +- 0.3   |   60.6 +- 0.6   |   89.7 +- 0.5   |   65.8 +- 0.4   |   71.4 +- 0.4   |
> | JumpStyle | **71.3 +- 0.3** | **66.6 +- 0.3** | **96.7 +- 0.2** | **68.1 +- 0.9** | **75.7 +- 0.4** |
>
> > Ablation studies, or other types of evidence, are needed regarding why using weak augmentation to obtain pseudo-label and strong autmentation for entropy minimization? If the weak augmentation is only taking examples with high label agreement, wouldn’t it somewhat equivalent to entropy minimization, i.e., making the prediction more confident if it is already robust?
>
> In this work, we check for the consistency between the predictions of the original samples and its weak augmentation to obtain the pseudo labels, inspired from the Semi-Supervised learning frameworks [1][2]. Compared to using pseudo labeling solely based on the samples, this results in better quality pseudo labels as we discard samples which flip the predictions for minor perturbations to the sample. We apply pseudo label loss for strong style augmentations.
> In addition to entropy minimization as in [3] that drives the model to provide confident predictions.
> pseudo labeling  provides stronger supervision to cluster the test domain samples and better regularizes the model as we are enforcing the model to provide consistent predictions for strong perturbations as well. To analyze the importance of the two loss components, we experiment on the VLCS dataset, where we only use the entropy loss with strong augmentations to compare with JumpStyle, where both entropy loss and pseudo label loss is used on strong augmentations. The following results show that incorporating pseudo label loss in addition to entropy loss provides a 2% gain in accuracy. As we use the strong augmentations for pseudo label loss, we choose to use the same for entropy loss for efficient back propagation. On performing experiments where we use weak augmentations for entropy loss as an alternative to strong augmentations in Jumpstyle, we observe similar results (differing by only 0.2%) as shown in Table below. Hence, we simply use the strong augmentations to compute both the loss components.
>
> | Method                                             | VLCS        |
> |----------------------------------------------------|-------------|
> | Entropy loss with strong augmentations             | 73.5 +- 0.5 |
> | JumpStyle with weak augmentations for entropy loss | 75.5 +- 0.4 |
> | JumpStyle                                          | 75.7 +- 0.4 |
>
> [1] Kihyuk Sohn, David Berthelot, Nicholas Carlini, Zizhao Zhang, Han Zhang, Colin A Raffel, Ekin Dogus Cubuk, Alexey Kurakin, and Chun-Liang Li. Fixmatch: Simplifying semi-supervised learning with con- sistency and confidence. NeurIPS, 2020.
>
> [2] D. Berthelot, N. Carlini, I. Goodfellow, N. Papernot, A. Oliver, and C. A Raffel. Mixmatch: A holistic approach to semi-supervised learning. NeurIPS, 2019.
>
> [3] D. Wang, E. Shelhamer, S. Liu, B. Olshausen, and T. Darrell. Tent: Fully test-time adaptation by entropy minimization. In ICLR, 2021.

---

> > ### Author Response · Authors · 2022-12-20
> > **Response to Reviewer 184g**
> >
> > Dear reviewer,
> >
> > Thank you for the valuable feedback! Please find the detailed answers for the requested questions below:
> >
> > > Ablation studies are needed to justify the need for style-augmentation over other types of input augmentation. The authors should also clarify the difference between styles and domains, and how style augmentation within the same test domain could be helpful.
> >
> > Though the standard augmentations (like rotation, crop, etc), etc. in general help to improve the test performance (Table 3), they are very generic and may not be the best for our application. The ablation study on input augmentations has been done in Table 3. Here, we compare Jumpstyle (style based augmentation) with the augmentations used in Fixmatch (crop, rotate as weak augmentation and color, auto contrast, shear,  brightness,  solar with high intensity as strong augmentation).
> >
> > Although the style augmentations we use are inspired from MixStyle, our objectives are very different. For domain generalization, during training, the characteristics from samples of  different domains are mixed to synthesize pseudo-domains, so that the model learns a domain invariant representation and generalizes well  during testing. .
> >
> > In our work, all the test data belong to a single domain. But the styles of the images from this domain can be distinctive from the source domains used for training the model. In addition, style is specific for an instance of the image. Our motive is more inclined to single image style transfer where the style of one image is transferred to another. For eg., given two paintings, we want to transfer the style of one painting to another.  Here, both the images come from the same domain, Painting. In the current TTA setting, though we have samples from the same domain, as we are learning in an online fashion, using the current batch to mix styles shows the model additional variations within the test domain. Enforcing consistent predictions across such augmentations helps the model learn finer variations of the test domain as compared to using the standard augmentations This helps the model to be more robust and hence generalizing better for the future test batches.
> >
> > > In Sec. 6.3 (3), it is unclear why alpha is a constant hyper-parameter instead of a function over batch size. The authors should also clarify how alpha is configured in the ablation with different batch sizes in Table 2.
> >
> > In our work, alpha is a function of batch size and is computed as in  eq.(3).  In Table 2, alpha is calculated using eq.(3) for different batch sizes.  In Table 6 also, the alpha for our method (0.6) was computed using eq.(3) as a function of batch size, which was 32.

---

> > ### Comment · Reviewer_184g · 2023-01-02
> > **Thanks for the response**
> >
> > I'd like to thank the authors for the response and the additional ablation studies. I'd like to seek further clarification on the problem setup.
> >
> > > The test-time adaptation setting for the domain generalization set-up is as follows: Training Stage: The model is first trained using multiple source domains as in the standard domain generalization scenario. Testing Stage: Given only the trained model, it is adapted in an online manner where the test samples come from an unseen test domain which is different from the source domains seen during training.
> >
> > Could you clarify in math terms what is "online manner" exactly? How it differs from [transductive machine learning](https://en.wikipedia.org/wiki/Transduction_(machine_learning)). What makes the problem setup "online"?

---

> > > ### Author Response · Authors · 2023-01-03
> > > **Response to Reviewer 184g**
> > >
> > >
> > > In transductive machine learning, the the model has access to both the labeled and unlabeled samples for learning.
> > >
> > > In contrast, in our case, during the training stage, the model has access to only the data from the source domains, say $D_1, D_2,...,D_{d_{tr}}$. The labelled source domain samples ($x_{i}^{d}$, $y_{i}^{d}$)$\\sim D_k, k \in {1,2,...,d_{tr}}$ are used to train the model $F_{\theta}$ parameterised by  $\theta$ by optimising a training loss like Empirical Risk Minimisation (or any Domain Generalisation method) as follows:
> > >
> > >
> > > min $L_{ERM}$  =  $\\frac{1}{d_{tr}}$  $\sum_{d=1}^{d_{t r}}$ $ \frac{1}{n_{d}}$ $\sum_{i=1}^{n_{d}}$ $L_{CE}$($x_{i}^{d}$, $y_{i}^{d}$)
> > >
> > >
> > > This trained model $F_{\theta}$ would then be deployed in a test environment. For example, a vendor has trained an image classification model using several source domain data, but does not have information about the test domain and the test samples that will be used for testing by the client.
> > >
> > > Once the model is deployed for testing, the objective of Test time adaptation is to leverage the unlabelled test samples $x_t \sim D_{test}$ available at a time instant $t$ to adapt the given model $\theta$. The setup is called online because during testing, the entire unlabelled test set is not available at once. At time instant $t$, the model gets access to the test batch $x_t$ which is used to adapt the Batch Norm affine  parameters ${\gamma, \beta}$ using a test time loss, which for JumpStyle is as follows:
> > >
> > >  $min$ $L_{test}$ = $\\eta*L_{pl} + (1-\\eta)*L_{ent}$.
> > >
> > > The predictions for $x_t$ are made at the same time step $t$. This process is repeated for every test batch (which we refer to as online manner). At a time step $t+1$, the model no longer has access to $x_t$. We briefly describe the test time adaptation protocol below:
> > >
> > > >While True: #As long as test batches are coming online
> > >         > - Get a test batch $x_t$ from an unseen test domain $D_{test}$ at time instant t.
> > >         > - Update the BN affine parameters ${\gamma, \beta}$ by minimising $L_{test}(x_t)$ .
> > >         > - Evaluate the current batch $x_t$.

---

### Review · Reviewer_YCbc · 2022-12-06

**Summary Of Contributions:**

This paper proposes a jumpstyle test-time domain adaptation method. It proposes to reestimate batchnorm statistics based on the number of test samples available online, and reformulate mixup in the feature space to obtain local manifolds with different level of augmentations. Results on small scale/medium datasets shows its effectiveness.

**Audience:**

Yes

**Broader Impact Concerns:**

This paper does not introduce any concerns on the ethical implications. Author[s] are still encouraged to add such a section to discuss the paper.

**Claims And Evidence:**

Yes

**Requested Changes:**

1. Results on larger datasets following https://github.com/facebookresearch/DomainBed.
2. Give a clear explanation how jumpstyle can achieve excellent performance for unseen domain examples during testing using Eq. 2- 5.




**Strengths And Weaknesses:**

+:
1. Correcting test batch statistics by substituting the training domain information as a prior to improve the domain shift is technically sound.
2. The proposed method is simple and can be easily integrated into different domain generalization pipelines.

-:
"Generating pseudo-domains/styles helps to achieve excellent performance for unseen domain examples during testing." this somehow sounds not that plausible, in Eq. 5 and Eq. 4 how do you guarantee that the pseudo-domains can well aligned with the unseen test data? I am actually wondering about Eq. 1 as mentioned in the paper: How to generalize well if \mu_{s} and \sigma_{s} is quite different from \mu_{t} and \sigma_{t}? Tent sounds like pretraining the weights of BN modules with the training data and re-estimated with test data. More general, what's the key difference compared to simply pretraining on more diverse upstream data using all learnable layers and then conduct few-shot finetuning with some adaptable modules? Literature in this field also shows much stronger results compared to results in Table 1.

---

> ### Author Response · Authors · 2022-12-20
> **Response to Reviewer YCbc**
>
> Dear reviewer,
>
> Thank you for the valuable feedback! Please find the detailed answers for the requested questions below:
>
> > "Generating pseudo-domains/styles helps to achieve excellent performance for unseen domain examples during testing." this somehow sounds not that plausible, in Eq. 5 and Eq. 4 how do you guarantee that the pseudo-domains can well aligned with the unseen test data? I am actually wondering about Eq. 1 as mentioned in the paper: How to generalize well if \mu_{s} and \sigma_{s} is quite different from \mu_{t} and \sigma_{t}?
>
> Since it is a test time adaptation setting, the adaptation is done on the incoming test data itself (from an unseen domain) in an online manner. The main challenge is that the number of samples in a batch may be very less. The information of the confident samples in the form of pseudo-labels help cluster the unseen domain features better. As we get more and more target data, the performance gradually improves.
> In cases where \mu_{s} and \sigma_{s} are very different from \mu_{t} and \sigma_{t}, the improvement obtained doing TTA can be less, however it always performs better than the source model. For eg., on visually observing, the domain shifts in Terra-Incognita are severe when compared to other datasets like VLCS and PACS. As expected the gain in performance when compared to source is less for Terra-Incognita, but is quite significant for VLCS and PACS.
>
> > Tent sounds like pretraining the weights of BN modules with the training data and re-estimated with test data. More general, what's the key difference compared to simply pretraining on more diverse upstream data using all learnable layers and then conduct few-shot finetuning with some adaptable modules? Literature in this field also shows much stronger results compared to results in Table 1.
>
> Test-time adaptation (TTA) is more realistic and challenging as compared to few-shot finetuning. In TTA, given only the trained source model (and no source data), the goal is to adapt the model on the incoming  test samples in an online manner. No labeled data from the test domain is available to the model for updating its parameters before the testing stage. This is more challenging when compared to few-shot fine-tuning, as this scenario assumes access to few labeled samples from the test domain for updating the model parameters before the testing phase..
>
> In this work we experimentally showed that the proposed JumpStyle framework can be used to improve results on unseen test data for different Domain Generalization methods. To demonstrate this, in addition to the experiments done using ResNet-18 trained using ERM as shown in Table 1, we performed experiments using the MixStyle trained ResNet-18 backbone and reported the results in Table 2, which shows that TTA using JumpStyle indeed works across DG methods.
> In addition, as suggested by the reviewer, we analyze the effect of different numbers of training domains on the DG as well as TTA performance. Towards this goal, we perform an experiment on the VLCS dataset, where we select two and three domains for training and evaluate the source trained model(Without adaptation), TTA approaches Tent with JumpStart Initialization(Tent+Jump) and the proposed JumpStyle framework on the unseen domain Labelme. As expected, the model trained on three source domains performs significantly better compared to the one trained using two domains as it is better generalized. But, in both cases, performing online TTA further improves the performance. This shows that, even if a well generalized model can be learnt during training, TTA can always be incorporated to further boost the performance.
>
> |   Training domains  | Test domain | Without adaptation | Tent+Jump | JumpStyle |
> |:-------------------|:-----------:|:------------------:|:---------:|:---------:|
> | VOC, SUN09          |   LabelMe   |        59.5        |    62.8   |    63.3   |
> | VOC, SUN09, Caltech |   LabelMe   |        65.0        |    66.0   |    66.5   |

---

> ### Author Response · Authors · 2022-12-20
> **Response to Reviewer YCbc**
>
> Dear reviewer,
>
> Thank you for the valuable feedback! Please find the detailed answers for the requested questions below:
>
> > Results on larger datasets following https://github.com/facebookresearch/DomainBed.
>
> DomainBed has 7 datasets, namely CMNIST, RMNIST, VLCS, PACS, OfficeHome, Terra-Incognita and DomainNet (from small to large in terms of number of samples). In two of these datasets, CMNIST and RMNIST, the domains are synthetically generated. The other five demonstrate more realistic domain shifts including domains like Photo, Art, Sketch etc. Following our previous benchmark T3A, we evaluate over 4 datasets including VLCS, PACS, OfficeHome, Terra-Incognita, which are also a part of DomainBed.
>
> >Give a clear explanation how jumpstyle can achieve excellent performance for unseen domain examples during testing using Eq. 2- 5.
>
> Inspired by the effectiveness of correcting BatchNorm statistics to account for testing distribution shift [1], we propose the Jumpstart initialization of BN statistics in eq.(2), which combines the source and test statistics as a function of the batch size. Since larger batch sizes provide better estimates of the test statistics, we propose a custom convex weight which can be automatically computed based on the batch size in eq.(3). The effectiveness of this module is presented in Table 3 and Table 6 in the paper.
>
> As we only have limited data for online adaptation, it is natural to augment the samples. Though the standard augmentations (like rotation, crop, etc), etc. in general help to improve the test performance (Table 3), they are very generic and may not be the best for our application. In order to better adapt to the unseen test domain, which may have different styles compared to those in the source domains (eg. different styles of sketching a cat by different people), we propose to use style augmentations instead of generic augmentation for the DG task. Eq.(4) describes how the channelwise feature mean and variance of two samples are mixed to generate a style augmented feature, (both weak and strong as in Eq.(5)).
>
> Utilizing the consistency of the predictions of the weak style augmentation and the original sample along with the prediction confidences for generating confident pseudo-labels has been successfully used in several semi-supervised approaches. The stronger style augmentations are given these pseudo-labels from which we get the pseudo label loss as described in eq.(6). This strong supervision further helps the model to adapt to the unseen target domain using few examples.
>
> [1]Steffen Schneider, Evgenia Rusak, Luisa Eck, Oliver Bringmann, Wieland Brendel, and Matthias Bethge. Improving robustness against common corruptions by covariate shift adaptation. NeurIPS, 2020.

---

### Review · Reviewer_UjrS · 2022-12-06

**Summary Of Contributions:**

This paper deals with the test-time adaptation task in a domain generalization setting. That is, adapting a DG model using online test data to further improve the test-time adaptation/generalization performance. This paper proposes using a combination of source statistics and test statistics as new BN statistics. Moreover, it adopts style-aware augmentation based pseudo-labeling tricks to  refine the affine transformation parameters. Experiments on various domain generalization benchmarks show the effectiveness of proposed method.

**Audience:**

Yes

**Broader Impact Concerns:**

I think this work may not raise serious ethical concerns.

**Claims And Evidence:**

No

**Requested Changes:**

See the weakness part.

**Strengths And Weaknesses:**

Pros:

-The paper is well-written and easy to read.

-The method is simple and reasonable.

-The experiment results are good.

Cons:

-The contribution seems limited. This paper seems using a combination of previous tricks (although they may not be originally designed for test-time domain adaptation/generalization) to deal with the test-time adaptation/generalization problem. I didn't see any novel or significant contributions made by this paper.

-JumpStyle only updates the affine transformation parameters of BN layers. How about updating all the network parameters?

-Table 4 showed some success cases and some failure cases. Are there any patterns exhibited by different methods? Do the authors have any insights?

---

> ### Author Response · Authors · 2022-12-20
> **Response to Reviewer UjrS**
>
> Dear reviewer,
>
> Thank you for the valuable feedback! Please find the detailed answers for the requested questions below:
>
> > The contribution seems limited. This paper seems using a combination of previous tricks (although they may not be originally designed for test-time domain adaptation/generalization) to deal with the test-time adaptation/generalization problem. I didn't see any novel or significant contributions made by this paper.
>
> The major contribution of this work is to show that for various approaches which use batch norm in their architecture, whether they are the simple approaches or the state-of-the-art techniques trained for the domain generalization task, the results can be significantly improved using TTA setting.
> We agree with the reviewer that many of the concepts have been inspired by existing techniques. But there are several subtle, but important application specific modifications to the existing approaches, which together lead to the impressive performance of the proposed framework. Few of the contributions are:
> 1) Weak and strong style augmentations for consistency regularization [1] is one of the novel contributions which outperforms the standard image augmentations for this task. Here, we randomly mix the styles of samples within the batch from test domain (instead of the domains as in the original work [2]), to better adapt the model to the current domain, which may have different variations as compared to the seen data.
> 2) Using confident pseudo-labels based on our proposed augmentations help in test-time adaptation and improves the performance across batch sizes, which to the best of our knowledge has not been explored for the TTA setting.
> 3) Performs significantly better than the SOTA approach T3A [3] for DG task.
>
> > JumpStyle only updates the affine transformation parameters of BN layers. How about updating all the network parameters?
> As suggested by the reviewer, we performed an experiment on the VLCS dataset, where we updated all the parameters for two different batch sizes. The results for this experiment and comparison when only the BN parameters are updated are shown below.
>
> | Batch size | All params | BN params |
> |------------|----------------|---------------|
> | 16         | 75.60 +- 0.4   | **75.70 +- 0.4**  |
> | 32         | 74.90 +- 0.3   | **74.95 +- 0.2**  |
>
>
> From the above results, we observe that adapting all the parameters does not result in any additional gains in terms of accuracy. Also, since the data in each batch in an online setting is very less, it is more efficient to update a few parameters as compared to updating all the parameters.
>
> > Table 4 showed some success cases and some failure cases. Are there any patterns exhibited by different methods? Do the authors have any insights?
>
> As suggested, we analyzed several of the wrong class predictions for OfficeHome and VLCS datasets. We found that many mistakes of the proposed method are semantically logical. For example, for OfficeHome data, JumpStyle predicts the laptop as monitor and soda can as mug. For VLCS data, many of the wrong classifications are happening for images where multiple classes are present.
>
> [1] Kihyuk Sohn, David Berthelot, Nicholas Carlini, Zizhao Zhang, Han Zhang, Colin A Raffel, Ekin Dogus Cubuk, Alexey Kurakin, and Chun-Liang Li. Fixmatch: Simplifying semi-supervised learning with con- sistency and confidence. NeurIPS, 2020.
>
> [2] K. Zhou, Y. Yang, Y. Qiao, and T. Xiang. Domain generalization with mixstyle. In ICLR, 2021.
>
> [3] Yusuke Iwasawa and Yutaka Matsuo. Test-time classifier adjustment module for model-agnostic domain generalization. NeurIPS, 2021.

---

### Decision · Action_Editors · 2023-01-23

**Recommendation:** Reject

**Comment:**

This paper proposed a test-time adaptation algorithm for domain generalisation by improving batch normalisation and introducing style augmentations with pseudo-label. All three reviewers were leaning reject, due to their shared concerns about the audiences' potentially limited interest in the proposed minor technical changes and the insufficient experimental results.
it might be a limitation for the authors to only focus on batch normalisation and the Tent is the only baseline to try out the effectiveness of the proposed changes. It would be more interesting if the authors can enhance the techniques accordingly to investigate other normalisations (e.g., layer or instance normalisation) and include more baseline algorithms/backbones to evaluate the generic enough changes. The authors thus need a significant revision before considering a resubmission.

**Audience:**

Reviewers commented that the proposed algorithm strongly relies on those well-established works, which were admitted by the authors as well. The authors were well inspired by these existing works and conducted some application-specific modifications. But it seems these modifications have not been well studied in current experiments yet. The only relevant ablation studies were in Table 3 (which was not correctly cited or discussed in the main paper) and small tables in the response to Reviewers. A lack of impressive studies on the new modifications is a major issue that would hurt the audiences' interest in the proposed changes.

**Claims And Evidence:**

This paper proposed a test-time adaptation algorithm for domain generalisation by improving batch normalisation and introducing style augmentations with pseudo-label. The authors stated that the proposed algorithm can be "generalizable not only across different backbone". But in fact, the authors only evaluated the algorithm over ResNet-18 and ResNet-50 backbones, and the algorithm performance gain became more marginal along with the increasing of the network complexity, e.g., +1.5% on ResNet-50 and 2.4% on ResNet-18. Notably, T3A is a closely related work to this paper, and in T3A paper, it has been evaluated on many bigger networks, e.g., BiT-M-R50x3, BiT-M-R101x3,  etc. The overall evidence in this paper can thus be insufficient to support the authors' claims.